# DECOUPLED GREEDY LEARNING OF GRAPH NEURAL NETWORKS

## ABSTRACT

Graph Neural Networks (GNNs) become very popular for graph-related applications due to their superior performance. However, they have been shown to be computationally expensive in large scale settings, because their produced node embeddings have to be computed recursively, which scales exponentially with the number of layers. To address this issue, several sampling-based methods have recently been proposed to perform training on a subset of nodes while maintaining the fidelity of the trained model. In this work, we introduce a decoupled greedy learning method for GNNs (DGL-GNN) that, instead of sampling the input graph, decouples the GNN into smaller modules and associates each module with greedy auxiliary objectives. Our approach allows GNN layers to be updated during the training process without waiting for feedback from successor layers, thus making parallel GNN training possible. Our method achieves improved efficiency without significantly compromising model performances, which would be important for time or memory limited applications. Further, we propose a lazy-update scheme during training to further improve its efficiency. We empirically analyse our proposed DGL-GNN model, and demonstrate its effectiveness and superior efficiency through a range of experiments. Compared to the sampling-based acceleration, our model is more stable, and we do not have to trade-off between efficiency and accuracy. Finally, we note that while here we focus on comparing the decoupled approach as an alternative to other methods, it can also be regarded as complementary, for example, to sampling and other scalability-enhancing improvements of GNN training.

## 1 INTRODUCTION

Graph Neural Networks (GNN) have been shown to be highly effective in graph-related tasks, such as node classification (Kipf & Welling, 2016), graph classification (Ying et al., 2018b), graph matching (Bai et al., 2019), and recommender system (Ying et al., 2018a). Given a graph of arbitrary size and attributes, GNNs obtain informative node embeddings by first conducting a graph convolution operation to aggregate information from the neighbors of each node, and then transforming the aggregated information. As a result, GNNs can fuse together the topological structure and node features of a graph, and have thus became dominant models for graph-based applications.

Despite its superior representation power, the graph convolution operation has been shown to be expensive when GNNs become deep and wide (Chen et al., 2017). Therefore, training a deep GNN model is challenging for large and dense graphs. Since deep and wide GNNs are becoming increasingly important with the emergence of classification tasks on large graphs, such as the newly proposed OGB datasets (Hu et al., 2020), and semantic segmentation tasks as introduced in (Li et al., 2019), we focus here on studying methods for alleviating computational burdens associated with large-scale GNN training.

Several strategies have been proposed during the past years to alleviate this computation issue of large-scale GNNs. GraphSAGE (Hamilton et al., 2017) took the first step to leverage a neighborhood sampling strategy for GNNs training, which only aggregates a sampled subset of neighbors of each node in the graph convolution operation. However, though this sampling method helps reduce memory and time cost for shallow GNNs, it computes the representation of a node recursively, and the node's receptive field grows exponentially with the number of GNN layers, which may make

the memory and time cost even goes larger for deeper GNNs when the sample number is big. The work of Chen et al. (2017; 2018); Zou et al. (2019) developed sampling-based stochastic training methods to train GNNs more efficiently and avoid this exponential growth problem. Chiang et al. (2019) proposed a batch learning algorithm by exploiting the graph clustering structure. Beyond the aforementioned methods, recently, You et al. (2020) proposed a layer-wise sequential training algorithm for GNNs, which decouples the aggregation and transformation operations in the per-layer feed-forward process and reduces the time and memory cost during training while not sacrificing too much model capability, this indicates that the GNN layers do not have to be learned jointly. However, the sequential training would bring some inefficiency.

In addition to the inefficiency brought by the graph convolution operation, as discussed in (Belilovsky et al., 2019a), the sequential nature of standard backpropagation also leads to inefficiency. As pointed out in (Jaderberg et al., 2017), backpropagation for deep neural networks suffers an update-locking problem, which means each layer heavily relies on upper layers' feedback to update itself, and thus, it must wait for the information to propagate through the whole network before updating. This would be a great obstacle for GNN layers to be trained in parallel to alleviate computation pressure under time and memory constraint, and would prohibit the GNN training to be trained in an asynchronous setting.

In this work, using semi-supervised node classification as an example, we show that the greedy learning would help to decouple the optimization of each layer in GNNs and enable GNNs to achieve update-unlocking, i.e., allow the GNN layers to update without getting any feedback from the later layers. By using this decoupled greedy learning for GNNs, we can achieve parallelization of the network layers, which would make the model training much more efficient and would be very important for time or memory limited applications. Moreover, we propose to use a lazy-update scheme during training, which is to exchange information between layers after a certain number of epochs instead of every epoch, this will further improve the efficiency while not sacrificing much performance. We theoretically analyze the computation complexity of our proposed method, and analogue our method to the classic block coordinate descent optimization to enable further analysis. We run a set of experiments to justify our model, and show its great efficiency on all benchmark datasets. On the newly proposed large OGBN-arxiv dataset, when training a 7-layer model, our proposed method even saves 85% time and 66% per-GPU memory cost of the conventionally trained GCN.

Our main contributions can be summarized as follows. **First**, we introduce a decoupled greedy learning algorithm for GNNs that achieves update-unlocking and enables GNN layer to be trained in parallel. **Next**, we propose to leverage a lazy-update scheme to improve the training efficiency. We evaluate our proposed training strategy thoroughly on benchmark datasets, and demonstrate it has superior efficiency while not sacrificing much performance. **Finally**, our method is not limited to the GCN and the node classification task, but can be combined with other scalability-enhancing GNNs and can be applied to other graph-related tasks.

## 2 RELATED WORK

Before discussing our proposed approach, we review here related work on efficient training strategies for GNNs. The computational complexities the discussed methods are summarized in Table 1, and we refer the reader to Appendix A for detailed computation.

### 2.1 DEEP GRAPH CONVOLUTIONAL NETWORK (DEEPGCN)

Graph convolutional network (GCN, Kipf & Welling, 2016) is one of the most popular models for graph-related tasks. Given an undirected graph $\mathcal{G}$ with node feature matrix $\boldsymbol{X} \in \mathbb{R}^{N \times D}$ and adjacency matrix $\boldsymbol{A} \in \mathbb{R}^{N \times N}$ where $N$ is node number and $D$ is feature dimension, let $\tilde{\boldsymbol{A}} = \boldsymbol{A} + \boldsymbol{I}$ , $\tilde{\boldsymbol{D}}$ be a diagonal matrix satisfying $\tilde{\boldsymbol{D}}_{i,i} = \sum_{j=1}^{N} \tilde{\boldsymbol{A}}_{i,j}$, and $\boldsymbol{F} = \tilde{\boldsymbol{D}}^{-1/2} \tilde{\boldsymbol{A}} \tilde{\boldsymbol{D}}^{-1/2}$ be the normalized $\tilde{\boldsymbol{A}}$, then, the $l$-th GCN layer will have the output $\boldsymbol{H}^{(l)}$ as $\boldsymbol{H}^{(l)} = \sigma(\boldsymbol{F} \boldsymbol{H}^{(l-1)} \boldsymbol{W}^{(l)})$, where $\sigma$ is the non-linear transformation, and $\boldsymbol{W}^{(l)}$ is the trainable weight matrix at layer $l$.

As pointed out in Li et al. (2018), when GCN becomes deep, it will suffer severe over-smoothing problem, which mean the nodes will become not distinguishable after stacking too many network layers. However, for applications such as semantic segmentation (Li et al., 2019) or classification

Table 1: Summary of Complexity. Here $\bar{D}$ denotes the average degree, $b$ denotes the batch size, $s_{node}$ and $s_{layer}$ are the number of sampled neighbors in NS and IS respectively, $K$ is the dimension of embedding vectors (for simplicity, assume it is the same across all layers), $L$ is the number of layers, $N$ is the number of nodes in the graph, $\boldsymbol{A}$ is the adjacency matrix, $T$ is the number of iterations, $T_{wait}$ is the waiting time for LU-DGL-GCN.

| Methods | Memory (per GPU) | Time |
|---|---|---|
| Full-Batch GCN (Kipf & Welling, 2016) | $\mathcal{O}(LNK + LK^2)$ | $\mathcal{O}(TL\|\boldsymbol{A}\|_0 K + TLNK^2)$ |
| GraphSage (Hamilton et al., 2017) | $\mathcal{O}(bKs_{node}^{L-1} + LK^2)$ | $\mathcal{O}(bTKs_{node}^L + bTK^2 s_{node}^{L-1})$ |
| VR-GCN (Chen et al., 2017) | $\mathcal{O}(LNK + LK^2)$ | $\mathcal{O}(b\bar{D}TKs_{node}^{L-1} + bTK^2 s_{node}^{L-1})$ |
| FastGCN (Chen et al., 2018) | $\mathcal{O}(LKs_{layer} + LK^2)$ | $\mathcal{O}(TLKs_{layer}^2 + TLK^2 s_{layer})$ |
| LADIES (Zou et al., 2019) | $\mathcal{O}(LKs_{layer} + LK^2)$ | $\mathcal{O}(TLKs_{layer}^2 + TLK^2 s_{layer})$ |
| ClusterGCN (Chiang et al., 2019) | $\mathcal{O}(bLK + LK^2)$ | $\mathcal{O}(TL\|\boldsymbol{A}\|_0 K + TLNK^2)$ |
| L2GCN (You et al., 2020) | $\mathcal{O}(NK + 2K^2)$ | $\mathcal{O}(L\|\boldsymbol{A}\|_0 K + 2TLNK^2)$ |
| LU-DGL-GCN (ours) | $\mathcal{O}(NK + 2K^2)$ | $\mathcal{O}(T\|\boldsymbol{A}\|_0 K/T_{wait} + 2TNK^2)$ |

tasks on large datasets (Hu et al., 2020), we do need deeper GCN models. Therefore, we follow the work of Li et al. (2019), alleviating over-smoothing problem by adding residual links between GCN layers and obtain the deepGCN model. The $l$-th layer of our network model will be $\boldsymbol{H}^{(l)} = \sigma(\boldsymbol{F}\boldsymbol{H}^{(l-1)}\boldsymbol{W}^{(l)}) + \boldsymbol{H}^{(l-1)}$.

## 2.2 Efficient GNN Training

To alleviate the expensive computation issue of GNN introduced in previous section, a lot of literature has proposed sampling-based batch-learning algorithms to train GNNs more efficiently.

GraphSAGE (Hamilton et al., 2017) introduced a node sampling strategy (NS), which is to randomly sample $s$ neighbors for each node at each layer, then, for each node, instead of aggregating embeddings of all its neighbors, we only aggregate the sampled ones. VRGCN (Chen et al., 2017) also followed this NS strategy, but it further proposed to leverage history activation to reduce the variance of the estimator. Though NS scheme has smaller complexity compared to full-batch GNN, there exists redundant computation and the complexity grows exponentially with the layer number.

Layer-wise importance sampling strategy (IS) would be a more advanced method for efficient GNN training. FastGCN (Chen et al., 2018) proposed to sample nodes for each layer with a degree-based sampling probability in order to solve the scalability issue in NS. The work of LADIES (Zou et al., 2019) leveraged IS idea as well, but it proposed a layer-dependent importance sampling scheme, which enjoys a smaller variance while maintaining same level complexity as FastGCN. Though this IS is better than NS in general, but we may still have to trade-off the complexity with the performance (i.e. accuracy for classification tasks), since using a large sample number would be helpful for the performance and increase the computation cost and vice versa.

Except for ther aforementioned methods, we also has ClusterGCN (Chiang et al., 2019), which proposes to partition the graph into several clusters then randomly select multiple clusters to form a batch to train the GNN. Though this would allows us to train much deeper GCN without much time and memory overhead, the stability of the performance of this approach would be hard to guarantee, since the performance would heavily depends on the graph clustering settings.

## 2.3 Layer-wise GNN

Layerwise learning for neural networks is first introduced by the work of (Hinton et al., 2006), and was further discussed in (Bengio et al., 2007). The work of (Belilovsky et al., 2019a;b) explored the layerwise CNNs and achieved impressive results.

Recently, (You et al., 2020) proposed a layerwise algorithm for GNN training. The key idea is to train GNNs layer by layer sequentially. Figure 1 illustrates the sequential training framework for layerwise GNN. For a $L$-layer GNN, we first train its first layer with an auxiliary classifier, then after it get fully converged, we fix this layer, and start to optimize next layer, we do the same things for all $L$ layers. This layerwise training saves us a lot of memory, since this method only requires us

to focus on one layer and only need to store one layer's activation results. Besides the clear memory saving, this layerwise training scheme also saves us a lot of time. During the learning process, it can decouple the two key components in the per-layer feed-forward graph convolution: aggregation and transformation. Then for each layer, it only needs to conduct the aggregation once at the beginning of the training, and then only need to do the transformation step at each iteration, which greatly reduce the time cost. According to the reported results, this sequentially-trained layerwise method is efficient than the joint learning strategy and can get us good performance. However, its sequential scheme would bring some inefficiency because one layer has to wait until its previous layers to get fully converged to start training. In our work, we solve this problem and enable the model parallelization to further improve the effciency.

## 3 PROPOSED APPROACH

### 3.1 MODEL ARCHITECTURE

As mentioned in section 2.1, we introduce our proposed algorithm with deepGCN model (i.e., the GCN model with residual link) since the residual link would help to alleviate GCN's over-smoothing problem when it goes deep, which would be important for large scale scenarios.

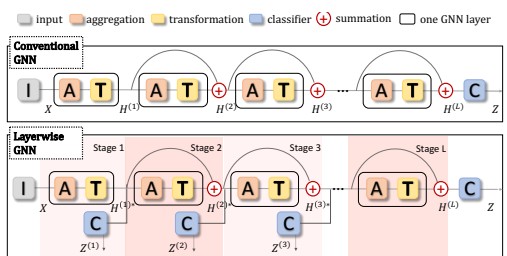

Figure 1: High level framework of conventional deepGCN (upper) and layerwise deep-GCN (lower). The aggregation step (A) corresponds to $FH^{(l-1)}$ operation and the transformation step corresponds to $\sigma(\cdot W^{(l)})$ operation. In conventional training, both steps are done for every iteration, but for sequential layerwise training, we can decouple these two operations to conduct the transformation step in every iteration, but the aggregation step only once at the beginning of each layer, which results in the demonstrated time saving.

There exist two ways to train such GCN model: conventional training and sequential layerwise training. We illustrate these two strategies with the high-level framework shown in Figure 1. For conventional training, we jointly optimize the learnable parameters in all layers and in the classifier. For layerwise training, we break the training for a $L$-layer GNN into $L$ sequential stages, each stage has to wait all its previous layers to get fully converged to start training.

Note that, the sequential layerwise training has the advantage that it can save time and memory while not compromising too much performance, this suggests its promising applications in large scale models under hardware and time constraints. We now consider, *whether we can extend it to a parallel version, so that the efficiency can be further improved?* Interestingly, as shown in the following sections, we find the answer is affirmative.

### 3.2 DECOUPLED GREEDY LEARNING ALGORITHM

To enable parallel GNN training, the most challenging problem is update-locking. Before updating one layer, we have to wait after the signal has been passed through all its successors, which would bring inefficiency. To alleviate this problem, we follow the design of layerwise GNN: decoupling the GNN model into different layers, associating each layer with an auxiliary classifier, which is a MLP layer with softmax activation, and assigning a per-layer greedy objective. Then, with the output activation of a given layer, we can leverage the auxiliary classifier to optimize the per-layer objective and therefore can update the current layer without any feedback from its successors while the rest lay-

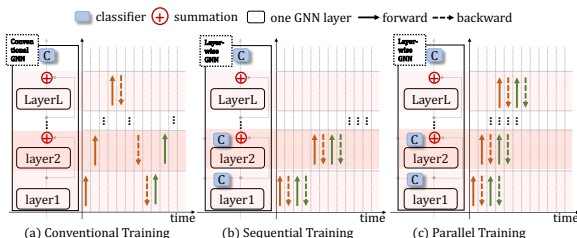

Figure 2: Signal propagation process for conventionally trained GNN, sequentially trained layerwise GNN, and the parallel trained GNN. Arrows of different colors represents different batches of data (can be either mini-batch or full-batch). We assume the forward process and backward process have the same time cost, we also assume the auxiliary classifier computation is negligible, these are only for simpler illustration purposes.

---

**Algorithm 1** Decoupled Greedy Learning (DGL) of GNNs

---

**Require:** Normalized Adjacency Matrix $\boldsymbol{F}$; Feature Matrix $\boldsymbol{X}$; Labels $\boldsymbol{Y}$; Total Number of Iterations $T$; Total Number of Layers $L$.

1: Initialize: $\boldsymbol{H}^{(0)} = \boldsymbol{X}$;
2: **for** $t = 1$ to $T$ **do**
3:     **for** $l = 1$ to $L$ **do**
4:         $\boldsymbol{H}^{(l)} = \sigma(\boldsymbol{F}\boldsymbol{H}^{(l-1)}\boldsymbol{W}^{(l)})$    // *Get node embeddings and store them as $\boldsymbol{H}^{(l)}$.*
5:         $(\boldsymbol{W}^{(l)}, \Theta^{(l)}) \leftarrow$ Update with $\nabla loss_{(\boldsymbol{W}^{(l)}, \Theta^{(l)})}(\boldsymbol{Y}, \boldsymbol{H}^{(l-1)}, \boldsymbol{F}; \boldsymbol{W}^{(l)}, \Theta^{(l)})$ // *Update parameters.*
6:     **end for**
7: **end for**

---

**Algorithm 2** Decoupled Greedy Learning (DGL) of GNNs with Lazy Update Scheme

---

**Require:** Normalized Adjacency Matrix $\boldsymbol{F}$; Feature Matrix $\boldsymbol{X}$; Labels $\boldsymbol{Y}$; Total Number of Iterations $T$; Total Number of Layers $L$; Waiting time $T_{lazy}$.

1: Initialize: $\hat{\boldsymbol{H}}^{(0)} = \boldsymbol{F}\boldsymbol{X}$;
2: **for** $t = 1$ to $T$ **do**
3:     **for** $l = 1$ to $L$ **do**
4:         $\boldsymbol{H}^{(l)} = \sigma(\hat{\boldsymbol{H}}^{(l-1)}\boldsymbol{W}^{(l)})$    // *Get node embeddings.*
5:         $(\boldsymbol{W}^{(l)}, \Theta^{(l)}) \leftarrow$ Update with $\nabla loss_{(\boldsymbol{W}^{(l)}, \Theta^{(l)})}(\boldsymbol{Y}, \hat{\boldsymbol{H}}^{(l-1)}; \boldsymbol{W}^{(l)}, \Theta^{(l)})$ // *Update parameters.*
6:         **if** $(t \bmod T_{lazy} == 0)$ **then**
7:             $\hat{\boldsymbol{H}}^{(l)} = \boldsymbol{F}\boldsymbol{H}^{(l)}$ // *Get propagated node embeddings and store them as $\hat{\boldsymbol{H}}^{(l)}$.*
8:         **end if**
9:     **end for**
10: **end for**

---

ers are still in the forward process. We name our training strategy as **D**ecoupled **G**reedy **L**earning of **GNN**s (DGL-GNN).

With our DGL-GNN, we achieve update-**un**locking, and therefore can enable parallel training for layerwise GNNs. For clarity, we provide Figure 2 to compare the signal propagation process of the conventionally trained GNN, sequentially trained layerwise GNN, and the parallel trained GNN. With this illustration, we can observe that the parallel training of layerwise GNN can avoid the case in which one layer is forwarding or back-propagating the signal while other layers are idle. Therefore, that given same number of batches of data, the parallel version would finish training much earlier than the conventional and the sequential version.

Following our notations in section 2, we denote by $\boldsymbol{F}$ the normalized adjacency matrix, $\boldsymbol{H}^{(l)}$ the output activation of $l$-th layer, and $\boldsymbol{W}^{(l)}$ the learnable parameters for $l$-th layer. Plus, let $\boldsymbol{Y}$ be the labels, $\Theta^{(l)}$ be the parameters for $l$-th layer's classifier, and $loss$ be the cross-entropy loss which is frequently used for classification tasks. Then, we have the per-layer objective function: $loss_{(\boldsymbol{W}^{(l)}, \Theta^{(l)})}(\boldsymbol{Y}, \boldsymbol{H}^{(l-1)}, \boldsymbol{F}; \boldsymbol{W}^{(l)}, \Theta^{(l)})$. We now formally define our DGL-GNN training method in algorithm 1. Note that, the inner for-loop can be done in a parallel manner, i.e., when the $l-$th layer is working on the backward process as given in line 5, the $(l+1)-$th layer can start forward propagation as given in line 4. Therefore, we claim our DGL-GNN algorithm can achieve update-**un**locking.

We then empirically observed that, without passing the signal to next layer immediately after the forward process, we still get same-level performance. Thus, we find that the efficiency of DGL-GNN can be further improved by leveraging an **L**azy **U**pdate scheme (LU-DGL-GNN). Instead of using the up-to-date activation output from its predecessor, one layer can use the history activation to learn its parameters and only update the history activation a few times during the overall training process. Then, same as sequential trained layerwise GNN, we only need to conduct the aggregation step for one time after each update, this saves us a lot time. We denote by $\hat{\boldsymbol{H}}^{(l)}$ the aggregated stored history activation for layer $l$. We now formally define the LU-DGL-GNN method in algorithm 2, we marked its difference with DGL-GNN in blue.

To sum-up, our proposed DGL-GNN and LU-DGL-GNN methods enjoys a very high efficiency because we introduce an auxiliary greedy objective for each layer and thus achieve update-**un**locking, we then decouple the model into layers and therefore enable the model to be trained in parallel, and we finally propose to leverage the lazy-update scheme, with which we can avoid redundant computation in the aggregation step and further reduce the training time.

## 4 COMPLEXITY ANALYSIS

As shown in Table 1, our proposed methods achieve a lower complexity compared to the conventional training and other baselines. Note that DGL-GCN can be regard as a special case in which $T_{wait} = 1$, i.e. we update the stored activation every epoch. Therefore, we focus on the complexity justification for LU-DGL-GCN.

For time complexity, first, we know that the training process consists two fundamental operations: aggregation and transformation. The time complexity of aggregation is $\mathcal{O}(\|\boldsymbol{A}\|_0 K)$, and the time complexity of the transformation step is $\mathcal{O}(NK^2)$. Then, we note that, for LU-DGL-GCN, during the full learning process, for each layer, we have to do aggregation $T/T_{wait}$ times, and we have to do the transformation $2T$ times because this step should be conducted for both the GNN layer and the auxiliary classifier. Since the computation for each layer is done in parallel, we know that the overall time complexity for LU-DGL-GCN should be $\mathcal{O}(T\|\boldsymbol{A}\|_0 K/T_{wait} + 2TNK^2)$. In practice, if we put different layers on different GPUs and do the training in parallel, there would be some extra non-negligible time cost for GPU communication.

For memory complexity, it also consists two components. We have to store two things for each layer during the training: the history activation and the intermediate learnable weight matrices for GNN and for the auxiliary classifier. The activation takes $\mathcal{O}(NK)$, and the two types of weight matrix take $\mathcal{O}(2K^2)$ space. Again, when we do the training in a parallel fashion and assign the layers to different machines, the per-GPU memory would only be $\mathcal{O}(NK + 2K^2)$, which is significantly reduced compare to most of the existing baselines.

## 5 ANALOGY TO BLOCK COORDINATE DESCENT

To justify the rationality of the proposed model, we present here an analogy of our decoupling approach to the classic Block Coordinate Descent (BCD) optimization strategy and its variants (Wright, 2015; Wu et al., 2008; Shi et al., 2016), which for completeness are discussed in Appendix A.2. We observe that, the sequential layerwise GNN share similar high-level idea with the BCD method. We regard each layer and its associated auxiliary classifier as a module. Then, for each module, all its parameters can be treated as a coordinate block, we order the coordinate block according to which layer it corresponds to. Note that, in BCD, for each iteration, we choose one coordinate block and optimize the overall training objective with respect to the chosen block. So if we keep choosing the first coordinate block until it fully converged, then keep choosing the second coordinate block, etc., until the last coordinate block fully converge, then this optimization process is the same as the learning process of a sequentially trained layerwise GNN.

We also oberve that, the DGL-GCN can be analog to the synchronous parallel BCD and the LU-DGL-GCN can be regard as an analogy of asynchronous parallel BCD. Note that our DGL-GCN and LU-DGL-GCN can be implemented in a parallel fashion, and their key difference is whether all the layers share a consistency and up-to-date information. Therefore, if we make the same analogy of learnable parameters and the coordinate blocks as in the above sequential version, then it would be easy to find the similarity between parallel BCD and our decoupled greedy learning methods. With such analogy, it would allow us to leverage existing theorems for BCD optimization to better understand and analyze the DGL-GCN and LU-DGL-GCN.

## 6 EMPIRICAL RESULTS

We evaluate our proposed algorithms with the multi-class node classification task. However, it should be noted that the decoupled greedy learning method can also be applied in other graph-related tasks and is not limited to node classification.

Table 2: Comparison of LU-DGL-GCN with baseline methods on benchmark datasets. For small datasets (Cora, Citeseer, Pubmed), we set the number of layers as 2, for large OGBN-arxiv dataset, we set it as 7. We set $T_{lazy} = 50$ here for LU-DGL-GCN. Results show that LU-DGL-GCN can achieve superior efficiency without much performance sacrifice.

| Dataset | Sample Method | Accuracy(%) | Total Time(s) | Mem(MiB) |
|---|---|---|---|---|
| Cora | Full-Batch GCN | $81.1 \pm 0.5$ | $4.8 \pm 0.4$ | 1275 |
|  | LADIES | $79.2 \pm 1.2$ | $3.1 \pm 0.6$ | 1249 |
|  | FastGCN | $78.9 \pm 3.2$ | $4.7 \pm 1.3$ | 1257 |
|  | LGCN | $78.9 \pm 0.8$ | $2.9 \pm 0.3$ | 1239 |
|  | LU-DGL-GCN | $79.3 \pm 1.0$ | $1.4 \pm 0.1$ | 1269 |
| Citeseer | Full-Batch GCN | $71.6 \pm 0.3$ | $5.6 \pm 0.2$ | 1401 |
|  | LADIES | $66.5 \pm 1.1$ | $2.4 \pm 0.3$ | 1327 |
|  | FastGCN | $62.9 \pm 2.3$ | $5.8 \pm 0.9$ | 1335 |
|  | LGCN | $69.7 \pm 0.7$ | $3.7 \pm 0.3$ | 1335 |
|  | LU-DGL-GCN | $68.8 \pm 0.8$ | $1.5 \pm 0.0$ | 1331 |
| Pubmed | Full-Batch GCN | $79.3 \pm 0.3$ | $10.8 \pm 0.2$ | 1445 |
|  | LADIES | $78.1 \pm 0.6$ | $2.7 \pm 0.3$ | 1333 |
|  | FastGCN | $42.6 \pm 3.4$ | $2.7 \pm 0.7$ | 1341 |
|  | LGCN | $77.8 \pm 0.7$ | $3.5 \pm 0.2$ | 1327 |
|  | LU-DGL-GCN | $77.2 \pm 0.5$ | $2.2 \pm 0.1$ | 1403 |
| OGBN-arxiv | Full-Batch GCN | $71.9 \pm 0.2$ | $162.9 \pm 23.4$ | 4327 |
|  | LADIES | $49.1 \pm 2.8$ | $50.4 \pm 16.8$ | 1301 |
|  | FastGCN | $21.6 \pm 0.0$ | $24.0 \pm 6.3$ | 1293 |
|  | LGCN | $68.8 \pm 0.1$ | $122.5 \pm 17.2$ | 1424 |
|  | LU-DGL-GCN | $69.3 \pm 0.3$ | $23.6 \pm 0.1$ | 1473 |

We use the following four public datasets for evaluation: cora, citeseer, pubmed (Sen et al., 2008), and OGBN-arxiv (Hu et al., 2020). We briefly introduce these datasets and summarize their statistic in Appendix A.3. We compare our method against several baseline models introduced in Section 2: Full-Batch GCN (Kipf & Welling, 2016), FastGCN (Chen et al., 2018), LADIES (Zou et al., 2019), and LGCN (You et al., 2020). For all the methods, we use the same deepGCN model architecture and set all the hidden-dimension as 128. We follow the public implementations of all the baselines, and use their parameter settings. We conduct the training for 10 times and take the mean and variance of the evaluation results. Further implementation details are provided in Appendix A.3.

We evaluate the performance of different methods with the following evaluation metrics: **Accuracy** (%): The micro F1-score of the test data at the convergence point. **Memory (MiB)**: The maximum per-GPU memory cost during training. **Total Running Time (s)**: The total training time (exclude validation) before convergence.

We summarize the classification performance results in Table 6, which demonstrates the efficacy of our approach. We can see that, for small-scale datasets such as Cora, Citeseer, Pubmed, our LU-DGL-GCN is the fastest among all the baselines, and its accuracy is only lower than Full-Batch GCN but higher than all the other efficient GCN training strategies. Since we only use a 2-layer GCN architecture for these small-scale datasets, we can not see much memory advantage. For the large-scale OGBN-arxiv dataset, we use a deeper 7-layer GCN architecture, then we can find that, compared to Full-Batch GCN, our LU-DGL-GCN is much faster and can save per-GPU memory while only sacrifice minor performance. Compared to sampling-base methods (LADIES and FastGCN), our LU-DGL-GCN has a better accuracy and is faster, while only need slightly more per-GPU memory. Note that for here, we set the sample number as 64 for FastGCN and LADIES, if we increase the sample number, we will obtain a higher accuracy but the time cost and memory cost would be larger. Compared to LGCN which is a sequantially-trained layerwise GCN, our LU-DGL-GCN has a clear advantage on the time cost, and can even obtain a little accuracy improvement. Our experimental results together with our analysis in previous sections indicate that our LU-DGL-GNN would be very helpful for time and memory limited applications, especially for the scenarios in which we need deep models. Furthermore, to establish the importance of different components of our method, we isolate certain aspects of it via the following ablation study.

Table 3: Comparison of LU-DGL-GCN with different $T_{lazy}$. Note that, $T_{lazy} = 1$ corresponds to the DGL-GNN model. Results show that with proper $T_{lazy}$, we can reduce time cost and improve accuracy while maintaining same memory cost.

| Dataset | Performance | $T_{lazy} = 1$ | $T_{lazy} = 5$ | $T_{lazy} = 10$ | $T_{lazy} = 20$ | $T_{lazy} = 50$ |
|---|---|---|---|---|---|---|
| Cora | Accuracy (%) | $78.9 \pm 0.8$ | $79.1 \pm 0.5$ | $79.0 \pm 0.1$ | $79.2 \pm 0.6$ | $79.3 \pm 1.0$ |
| | Total Time (s) | $1.7 \pm 0.1$ | $1.6 \pm 0.07$ | $1.6 \pm 0.2$ | $1.4 \pm 0.1$ | $1.4 \pm 0.1$ |
| | Memory (MiB) | 1269 | 1269 | 1269 | 1269 | 1269 |
| OGBN-arxiv | Accuracy (%) | $69.1 \pm 0.3$ | $69.5 \pm 0.2$ | $69.4 \pm 0.2$ | $69.3 \pm 0.3$ | $69.3 \pm 0.3$ |
| | Total Time (s) | $332.5 \pm 14.1$ | $85.8 \pm 1.4$ | $48.1 \pm 0.5$ | $33.1 \pm 0.1$ | $23.6 \pm 0.1$ |
| | Memory (MiB) | 1473 | 1473 | 1473 | 1473 | 1473 |

**Importance of Lazy Update Scheme**. We illustrate the advantage of the Lazy Update Scheme and show how the waiting time $T_{lazy}$ will influence the performance. We use the cora and OGBN-arxiv datasets as examples for small-size and large-size scenarios, and follow previous model architectures and parameter. We run the model for 200 epochs and compare the obtained accuracy, total running time, and memory. We summarize the results in Table 6. We find that, with the lazy update scheme, we can greatly reduce the time cost, and a proper waiting time $T_{lazy}$ would help to improve the accuracy a little bit since it can alleviate overfitting. In addition, when comparing the accuracy curves of different $T_{layer}$ shown in Figure 3, we find that large $T_{lazy}$ makes the training more stable.

**Sequential Training v.s. Parallel Training**. Finally, we briefly compare the sequential training and our parallel training of the greedy objective. Again, we use cora and OGBN-arxiv as examples and follow the above experiment settings. As shown in Figure 3, we find that in terms of accuracy, parallel training can quickly catch up with the sequential training, and is less likely to overfit.

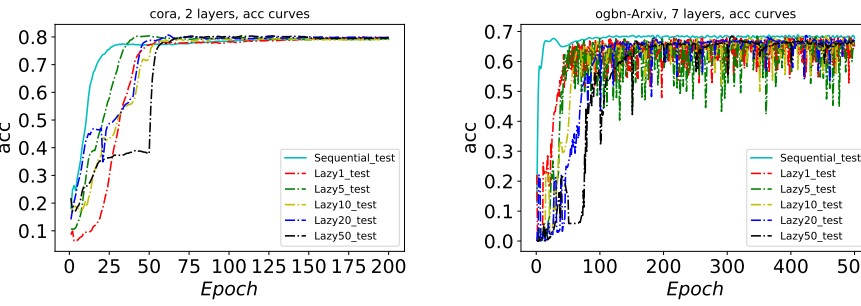

Figure 3: Comparison of sequential and parallel training. Results show that parallel can quickly catches up to sequential training within a few epochs.

## 7 CONCLUSIONS

In this paper, we focus on the efficiency issue of GNN training in large-scale applications and present a decoupled greedy GNN learning strategy. Our proposed DGL-GNN model achieves update-**un**locking by introducing greedy auxiliary objectives during training, and enables parallelization by decoupling the GNN into smaller modules. We also propose the LU-DGL-GNN method, which leverages a lazy-update scheme during training to further improve the model efficiency. We empirically analyze our proposed DGL-GNN model and demonstrate its effectiveness and superior efficiency through a range of experiments. We note that while we introduce our proposed method with deepGCN model and use the semi-supervised node classification task as an example, the DGL-GNN and LU-DGL-GNN methods are not limited to this setting, and can be applied to other GNNs and graph-related downstream tasks. Further, while here we focus on comparing the decoupled approach as an alternative to other sampling-based methods respect to their accuracy and efficiency, these approaches can be regarded as complementary to each other. By combining the decoupled greedy learning method with other scalability-enhancing improvements of GNN training, the computation cost would be further reduced, which poses a promising direction for future work.

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

# A    APPENDIX

## A.1    COMPLEXITY ANALYSIS FOR BASELINES

In this section, we explain how we compute the memory and time complexity in Table 1.

All the aforementioned methods's memory cost consists two parts: intermediate embedding matrices storage and weight matrices storage. The weight matrices always need $\mathcal{O}(LK^2)$ since it has to store $L$ weight matrices with dimension $K \times K$. The intermediate embedding has different memory cost for different methods. In terms of time complexity, it also consists two parts: the aggregation time cost and transformation time cost. These two parts varies for different methods.

Full-batch GCN stores all the intermediate embedding matrices for all the $L$ layers, and each matrice has $N$ nodes with dimension $K$, so its memory complexity for intermediate embedding storage would be $\mathcal{O}(LNK)$. Therefore, its total memory complexity is $\mathcal{O}(LNK + LK^2)$. In terms of time complexity, the propagation step which is a sparse-matrix multiplication has time complexity $\mathcal{O}(\|\boldsymbol{A}\|_0 K)$, and the transformation step which is a dense-matrix multiplication has time complexity $\mathcal{O}(NK^2)$. During the training for $L$ layers and $T$ iterations, the total time complexity would be $\mathcal{O}(TL\|\boldsymbol{A}\|_0 K + TLNK^2)$.

For GraphSAGE, it has to store $\mathcal{O}(bs_{node}^{L-1})$ $K$-dimension node embeddings, so it has a total memory complexity $\mathcal{O}(bKs_{node}^{L-1} + LK^2)$. In terms of time complexity, for each batch, to update one node, we have to update $\mathcal{O}(s_{node}^{L-1})$ activations, each needs $\mathcal{O}(\mathcal{K}\!\int_{\backslash \iota \lceil})$ for aggregation and $K^2$ for transformation. Therefore, the total time complexity for GraphSAGE is $\mathcal{O}(bTKs_{node}^{L} + bTK^2 s_{node}^{L-1})$.

For VR-GCN, it stores all historical activations, which takes a memory of $\mathcal{O}(LNK)$ and will lead to the total memory complexity $\mathcal{O}(LNK + LK^2)$. The time complexity can be analyzed similarly as GraphSAGE, which would be $\mathcal{O}(bT\bar{D}Ks_{node}^{L-1} + bK^2 s_{node}^{L-1})$.

For FastGCN, it only has to store $b$ node embeddings in the last layer, and $(L-1)s_{layer}$ node embeddings in the previous $L-1$ layers, so the total memory complexity would be $\mathcal{O}(bK + (L-1)Ks_{layer}) + LK^2)$. In terms of time complexity, we need $\mathcal{O}(bTKs_{layer} + (L-1)TKs_{layer}^2)$ for aggregation and $\mathcal{O}((bTK^2 + (L-1)Ts_{layer})K^2)$ for transformation. Note that, we have $b < s_{layer}$, so we ignore the relatively small terms and get the total memory complexity $\mathcal{O}(LKs_{layer} + LK^2)$, and total time complexity $\mathcal{O}(TLKs_{layer}^2 + TLK^2 s_{layer})$.

For LADIES, its is also a layer-wise sampling method. Though it has different sampling strategy as FastGCN, their memory cost and time cost are the same. There fore, LADIES also has memory complexity $\mathcal{O}(LKs_{layer} + LK^2)$ and time complexity and $\mathcal{O}(TLKs_{layer}^2 + TLK^2 s_{layer})$.

## A.2    SUPPLEMENTARY DISCUSSION OF COORDINATE AND BLOCK COORDINATE DESCENT

Coordinate descent (CD) is a classic iterative optimization algorithm that solves an optimization problem by approximately minimizing the objective along each coordinate directions successively.

Table 4: Statistics of Benchmark Dataset

| Dataset | Cora | Citeseer | Pubmed | OGBN-arxiv |
|---------|------|----------|--------|------------|
| Nodes | 2708 | 3327 | 19717 | 169343 |
| Edges | 5429 | 4732 | 44338 | 1166243 |
| Classes | 7 | 6 | 3 | 40 |
| Feature | 1433 | 3703 | 500 | 100 |

In each iteration, it would choose one variable, fix the other components, then optimizing the objective with respect to only the single variable. By doing so, we only need to solve a lower dimensional minimization problem at each iteration, which would be easier. CD algorithm has been discussed in various literatures and has been used in applications for a long time (Wright, 2015; Wu et al., 2008; Shi et al., 2016).

Block coordinate descent (BCD) is an extension of CD method. The difference of BCD and the conventional CD is that BCD will do the searching along a coordinate hyperplane instead of a single coordinate direction (Beck & Tetruashvili, 2013), i.e. it groups variables into blocks, and approximately minimizing the objective with respect to only one block of variables at each iteration while fixing the others.

For BCD algorithm, there exists its parallel implementations. As introduced in the work of (Wright, 2015), we can categorize them into two types: synchronous and asynchronous. For synchronous parallel BCD, we partition the computation into pieces and put different pieces on different processors, each processor will update a part of the variables in parallel, then a synchronization step should be conducted to guarantee the consistency of the information shared among all processors before further computation. For asynchronous setting, the difference is we do not have to do the synchronization. As discussed in Section 5, our proposed method can be regarded as analogous to BCD and its aforementioned variants.

### A.3    Supplementary for Experiments

**Dataset Statistics** We use four benchmark dataset: Cora, Citeseer, Pubmed, Reddit, and OGBN-arxiv for the node classification task. The detailed statictics of theses datasets are shown in Table 4.

**Hardware** We run the experiments on Tesla V100 GPU (16GB).

**Baselines** Detailed introduction of our baselines can be found in Section 2. As introduced before, as NS methods are in general less competitive than the IS methods, we only compare against the IS strategies.

**Model Architecture** For Cora, Citeseer, Pubmed, we set the number of layers as 2 for all the methods. For OGBN-arxiv, we set it to be 7.

**Parameter Settings** For all the methods and datasets, we conduct training for 10 times and record their mean variance. For small datasets (Cora, Citeseer, Pubmed), we set the epoch number as 200, for OGBN-arxiv, we set the epoch number as 500. We choose the model with the best validation performance as convergence point. For the sampling-based methods: FastGCN and LADIES, we set the sample number as 64, increasing this number would improve the accuracy a little bit, but would increase the memory and time cost.

