# OpenReview forum: "Decoupled Greedy Learning of Graph Neural Networks"
_ICLR.cc/2021/Conference — Reject_

### Official Review · AnonReviewer4 · 2020-10-13

**Rating:** 4
**Confidence:** 4

**Review:**

The paper gives up end-to-end training and instead trains graph convolution nets layer by layer. The authors call such layer-by-layer training process decoupled greedy learning (DGL-GNN).  In the decoupled greedy learning process, a lazy-update scheme is adopted, that is to exchange information between layers after a certain number of epochs instead of every epoch. The aggregation and transformation process are also separated and performed respectively.  Authors show their proposed engineering methods can save a constant fraction of computation time for graph convolution nets, but for the cost of sacrificing some predictive performance. Authors conduct some experiments on semi-supervised node classification data, reporting less memory and runtime but worse performance.

While the paper attempts to study an important problem of scaling graph networks to large graphs, and authors show the proposed methods is capable of saving in terms of memory and runtime by sacrificing predictive performance (which is not surprising at all), it is not suitable for publication at ICLR.

Comments:

Novelty is clearly below the bar of ICLR. The proposed method "layer-by-layer training" is rather trivial.

No theoretical or empirical justification is given for giving up end-to-end training. In fact, there are many easy counter-examples to show this approach would fail. Moreover, another problem with this method is the lack of auxiliary labels for each layer. It is easy to find problems where we do not have such layer wise auxiliary labels, and the proposed method would not work.

Moreover, even by doing layer by layer training, the proposed method cannot solve the scaling issue for billion scale networks, so some sampling is still needed.

This paper is purely experimental; however, the experiments are unsatisfactory:
- The reported performance is not competitive. In Table 2 and Table 3, test performance of proposed method is worse than prior works, e.g., LGCN, despite consuming even more memory cost. In this sense, practitioners would simply prefer LGCN to the layer-by-layer training.
- The datasets in experiments are very small, e.g., cora only have a few thousand nodes. What is the value of acceleration (while sacrificing performance) on such small graphs?
-  More graph neural network architectures need to be tested to show the effectiveness of the method: 1. graph attention network, 2. graph isomorphism network, 3. graphsage, 4. chebyshev GCN.
-  Missing link prediction benchmarks.
- Missing baselines: SIGN, GraphSAINT.


Grammar mistakes:
then after it get fully converged -> gets

optimize next layer -> the next

we do the same things -> thing

which scales exponentially with the number of layers -> factual error

---

> ### Author Response · Authors · 2020-11-24
> **Response to Reviewer4**
>
> Thank you for your comments. We address your questions and concerns in the following:
>
> 1. First of all, we'd like to reclaim our key contributions and novelty. In our work, we aim at designing algorithms to parallelize layer-wise training on GNNs to further enhance its efficiency, which would be extremely critical for deeper GNNs.
>
>
> 2. In terms of the theoretical justification, we try to make an analogy to the block coordinate descent algorithm to justify the rationality of our method. Another direction to provide some theoretical analysis would be considering its convergence guarantee. Inspired by the work of [1], we might be able to provide a convergence guarantee for every layer so the DGL method would converge to a local optimal. We thank the reviewer for pointing out our shortcomings in the theoretical part, and we would improve this part in the future version.
>
>
> Also, we do not require auxiliary labels, the auxiliary greedy learning objective added to each layer is a node classification objective, and we optimize this objective with the same training, validation, test set as the downstream node classification task.
>
>
> 3. As for the experiments:
>
> (1) We have tried to combine the proposed decoupled greedy learning method with the GIN model[2] on the node classification tasks on Cora and Pubmed. We set the number of layers to be 5, and set all the other hyper-parameters the same as in our previous experiments. We show our results in the following:
>
> Dataset | Cora | Pubmed |
>
> Metric | acc | time | acc | time |
>
> GIN | 75.70 ± 1.10 | 6.17 ± 0.38 | 73.49 ± 1.45 | 6.75 ± 0.31 |
>
> DGL-GIN | 77.02 ± 1.60 | 2.37 ± 0.04 | 74.26 ± 1.51 | 1.79 ± 0.01 |
>
> According to these results, we can find that, when combining with GIN, our proposed DGL method can achieve even better performance while only needing a significantly smaller time cost. (As for memory cost, since GIN only has one more parameter than GCN, their memory cost should be the same, so we can have the same conclusion that the per-GPU memory cost can be reduced.)
>
> (2) We have tried to test our proposed method with the GCN model and graph classification tasks on the MUTAG, PROTEINS, NCI1 datasets. We set the number of layers to be 3, and use a mean aggregator in the end to obtain global representation for each graph. We run the experiments 10 times and record the mean. According to our preliminary results, GCN obtained accuracy 77.37%, 72.32%, 70.01% on MUTAG, PROTEINS, NCI1 respectively, DGL-GCN obtained accuracy of 76.42%, 71.51%, and 67.63% respectively, while DGL-GCN only needs around half of the time and memory cost required by GCN.
>
> (3) We have tried to combine the proposed decoupled greedy learning method with the sampling-based FastGCN model on the node classification tasks on Cora, Citeseer, and Pubmed. We set the number of layers to be 5, and set all the other hyper-parameters the same as in our previous experiments. Different from our previous experiment, we only record the memory cost for the model and the intermediate activation, so that the per-GPU memory reduction can be found more clearly. We run the experiments for 200 epochs, and regard the model with the optimal validation performance as the best model and record its result on the test set. For each model and each dataset, we take the mean and variance of 10 runs. We set the number of sampled nodes as 512.
>
> Dataset | Cora | Citeseer | Pubmed |
>
> Metric | acc | time | acc | time | acc | time |
>
> FastGCN | 76.44 ± 1.30 | 4.51 ± 0.25 | 50.94 ± 2.64 | 5.09 ± 0.39 | 64.28 ± 1.06 | 4.89 ± 0.25 |
>
>
> DGL-FastGCN | 79.58 ± 1.05 | 1.40 ± 0.02 | 57.23 ± 3.85 | 1.46 ± 0.03 | 75.03 ± 1.28 | 1.63 ± 0.01 |
>
> According to our result, we can find that our proposed DGL method can be complementary to the sampling-based method, and can make those methods even more efficient.
>
>
>
> References:
>
> [1] Belilovsky, Eugene, Michael Eickenberg, and Edouard Oyallon. "Decoupled greedy learning of cnns." arXiv preprint arXiv:1901.08164 (2019).
>
> [2] Xu, Keyulu, et al. "How powerful are graph neural networks?." arXiv preprint arXiv:1810.00826 (2018).

---

### Official Review · AnonReviewer3 · 2020-10-26
**The greedy layer-wise decomposition of GNN in this paper is efficient in memory and time but the optimality of solution is of concern**

**Rating:** 6
**Confidence:** 5

**Review:**

GNN is known to be computationally complicated and it requires significant amount of time and memory. Unlike CNN, inference of the node embedding in GNN requires information be propagated throughout entire graph for each epoch. In this paper, the authors proposed to decouple the inference and aggregation in each layer and learn the layer-wise latent representation separately. The result is to decompose the deep GNN into a series of shadow networks that are connected sequentially. Since the optimization is done independently, it requires significant less time and memory.

This paper is clearly written and the contribution, literature review is sufficient for reader to understand. The analogy to block coordinate descent is a good part for reader to connect this work with the well-established work in optimization theory. The idea of layerwise decomposition is published in [You et al., 2020] and this paper improve over that by introducing more parallelization.

Pros:
1. The layer-wise decomposition is not a new idea but to optimize GNN, it is not as popular as in normal DL framework. The study on this topic will definitely brings some attention to the community and is enouraged.
2. The parallelization strategy and lazy update scheme is useful and easy to implement
3. In the experiment, it shows that the time reduction of LU-DGL-GCN is significant and it does not depends the depth of model. Therefore it is applicable to larger graph and deep structure.

Cons:
1.  Unlike the traditional convex optimization where the block coordinate decent is know to converge to optimal point. In the settings of GCN, there is no proof on how close the block coordinate descent would be towards to traditional GCN.
2.  Following the above question, it is known that the message-passing and aggregation in one layer only accumulate the information in 1-hop neighborhood and as a result, the embedding it learned only encodes the local structure within the 1-hop neighborhood. In many applications, the structural information of the graph can only be obtained through a wider p-hop neighborhood where p > 1. Traditional GNN take inference on p-hop neighborhood through  optimization of p-layers all together in backpropagation.  While the layer decomposition is attractive in time, LU-DGL-GCN loss the accuracy to infer the structural information in larger neighborhood. And because each layer is optimized using independent classifier on the same target, it is not hard to expect the node representation learned in the bottom layer to be less representative in LU-DGL-GCN compared to the traditional GCN as it does not encode the information beyond the 1-hop neighborhood.
3.  In early DL, the layer-wise optimization only used as pre-training to obtain a good weight initialization, and after that, a further full training is needed to obtain a representative embedding and model. It is suggested to conduct a full training with weight learned by LU-DGL-GCN and to compare with traditional GCN and to see if there are better solutions.
4.  It is better to discuss the limitation of the decoupled greedy learning, i.e. the model is myopic in that the resulting layer-wise node embedding only collect information in nearest neighbors. This is esp. an issue for lower-level representation since it lacks of prior learned from upper layer to guide its inference.
5.  The LU-DGL-GCN is a cascade  of  L independent shadow network models as opposed to one L-layer deep model. It is known that the representative and expressive power of the deep models is higher than that of a sequential connected independent shadow models. Thus compared to LU-DGL-GCN, it sacrifices the representation accuracy and model complexity to obtain a faster training and less memory. This is the tradeoff.

---

> ### Author Response · Authors · 2020-11-24
> **Response to Reviewer3**
>
> Thank you so much for your support and your valuable reviews. We address your main concerns in the following:
>
> 1) (for point 1) First of all, we analog our proposed decoupled greedy learning method to justify the rationality of the proposed method and to make things more clear, but they are not exactly the same. In terms of the convergence, though there is no theoretical proof on how close the block coordinate descent would be towards traditional GCN, our empirical results indicate that we would not sacrifice too much performance to achieve a much smaller time cost and per-GPU memory cost.
>
> 2) (for point 2,4,5) We do have p-hop information due to depth, just like any GCN. In the layer-wise training (both sequential and parallel version), the input of the p-th layer is still the output of the (p-1)-th layer, so for the p-th layer, a node is still able to get information from its p-hop neighbor (i.e. 1st layer can reach 1-hop neighbors, 2nd layer aggregates the embeddings of 1st layer so it can reach 2-hop neighbors, 3rd layer aggregates the embeddings of 2nd layer so it can reach 3-hop neighbors, …).
>
> 3) (for point 3,4) We agree that by adding further full training, the performance would be improved a little because in our current decoupled greedy learning method, each layer only leverages signals forwarded from its previous layers but does not get feedback from later layers. We thank the reviewer for pointing it out.

---

### Official Review · AnonReviewer1 · 2020-11-02
**Well written paper with some interesting ideas. Impact and experiments can be strengthened. Otherwise paper seems of lukewarm interest.**

**Rating:** 4
**Confidence:** 4

**Review:**

**Summary**
This paper introduces a method for decoupled layer wise training of graph neural networks. This method has the potential to be parallelized and offers computational and memory savings. The authors provide experimental results on 4 datasets and compare against 4 baselines. Results suggest that the approach achieves comparable results to other methods while being faster by parallelization.

**Strengths**
a) Well written paper with clear notation and illustrations

b) Good analysis of the different algorithms. Lays out complexity and memory costs of related work.

c) Analogy to block coordinate descent intuitively makes sense.

d) Nice contribution towards achieving asynchronous/lazy updates for GNNs.

**Weaknesses**

*Novelty and impact need to be strengthened*
a) As mentioned by the authors, greedy layerwise pre-training is an old idea. When it first came out for Deep NNs, everyone was excited about it but nowadays hardly anyone does it. So if this method has to make a comeback for GNNs then the benefits have to be very compelling.

b) Wrt benefits, the presentation around why this idea is compelling needs to be concretely laid out. It seems like there are some computational and memory benefits via parallelization. But is the additional complexity of parallel training worth it? As the authors mention, there can be non-trivial communication costs between GPUs in addition to the additional code complexity.

c) Absent theoretical convergence guarantees, the proposed approach is a heuristic as it relies on an auxiliary function that’s somewhat arbitrary. Further discussion is needed on why the chosen auxiliary function is a good idea. Some ablation studies with different auxiliary functions are needed to shed light on this particular choice of auxiliary function.

*Experiments can be strengthened*
a) The results seem to trade off accuracy with speed and saved memory and results are reported on 3 small and one large dataset. To truly claim that this method generalizes, the authors would need to strengthen their results across some different tasks (not just semi-supervised classification)  and models (not just GCN).

b) More details are needed regarding the experimental setup. Was there a multi-gpu setup? E.g. OGBN-arxiv has 7 layers. The authors report a total running time of 23.6s. Was this on a 7 GPU setup?

c) More ablation and convergence studies. The authors stop at 200 epochs for the smaller datasets. Does the proposed method reach full-batch GCN accuracy at higher epochs?

**Typos and other fixes**
a) Typo abstract: Should be “ Graph Neural Networks (GNNs) *have* become”

b) Table 6 is missing.

c) Results: “... but higher than all the other efficient GCN training strategies…” ; LGCN is better for pubmed and citeseer

d) “....when comparing the accuracy curves of different Tlayer shown in Figure 3, we find that large Tlazy makes the training more stable…”
i) Hard to see this from Figure 3. Consider plotting running variance as a shaded overlay.
ii) Plots are too busy. Hard to draw conclusions.


**What can the authors do to make the paper better**
a) More thorough experimentation as described in the *Experiments can be strengthened* section.

b) I did not find the code release with the paper. For a paper whose primary claim is computational and memory savings, I think it would be a good idea to include it.

c) Emphasize the novelty of the method otherwise, greedy pre training seems like a lukewarm idea.

---

> ### Author Response · Authors · 2020-11-24
> **Response to Reviewer1**
>
> Thank you so much for your constructive and valuable feedback. We address your questions and the main concerns in the following:
>
> 1) novelty and impact:
>
> Our proposed DGL-GNN method can improve the training efficiency of GNNs without significantly compromising model performances. By using the greedy layerwise training strategy, we would be able to enable parallelization of GNNs, which would be extremely helpful under time or per-GPU memory constraints.
>
> The layer-wise training is not a new research topic, but recently it starts to re-obtain attention from the machine learning community. The work of [1] explored its application in the large-scale image classifications task and found the layerwise training enjoys the benefit of easier theoretical analysis (e.g. convergence analysis), better inner organization, and better scalability for applications under memory constraints. The work of [2] explored its application in the graph domain, and found that except for the aforementioned benefits, the layerwise training can also help us to save more training time when combining with GNNs. In our work, we aim at designing algorithms to parallelize layer-wise training on GNNs to further enhance its efficiency, which is extremely critical for deeper GNNs.
>
> Our parallelization belongs to the model-parallelization scheme, where the resources (e.g., GPUs) are distributed according to the model components (i.e., layers in our case). Even if there might be some extra communication costs between GPUs, the overall time cost would be greatly reduced, and the memory burden would be distributed among multiple GPUs so the per-GPU memory cost is much smaller than the conventional joint training.
>
> The greedy objective is not an arbitrary one. In our experiment, we append the same classifier as the downstream task after each GNN layer, i.e., the auxiliary greedy learning objective is a node classification objective (with the same training, validation, test set as the downstream node classification task). With this simple but effective greedy objective, we encourage each layer to learn informative embeddings that can distinguish nodes from different classes. Exploring other or additional auxiliary functions (e.g., via self-supervised tasks) would be interesting, but that is orthogonal to our work.
>
> 2) experiments:
>
> We have tried to combine the proposed decoupled greedy learning method with the GIN model[3] on the node classification tasks on Cora and Pubmed. We set the number of layers to be 5, and set all the other hyper-parameters the same as in our previous experiments. We show our results in the following:
>
> Dataset    |                    Cora                   |                Pubmed               |
>
> Metric      |         acc         |       time       |         acc        |       time      |
>
>  GIN          | 75.70 ± 1.10 | 6.17 ± 0.38 | 73.49 ± 1.45 | 6.75 ± 0.31 |
>
>  DGL-GIN | 77.02 ± 1.60 | 2.37 ± 0.04 | 74.26 ± 1.51 | 1.79 ± 0.01 |
>
> According to these results, we can find that, when combining with GIN, our proposed DGL method can achieve even better performance while only needing a significantly smaller time cost. (As for memory cost, since GIN only has one more parameter than GCN, their memory cost should be the same, so we can have the same conclusion that the per-GPU memory cost can be reduced.)
>
> We have also tried to test our proposed method with GCN model and graph classification tasks on the MUTAG, PROTEINS, NCI1 dataset. We set the number of layers to be 3, and use a mean aggregator in the end to obtain global representation for each graph. We run the experiments 10 times and record the mean. According to our preliminary results, GCN obtained accuracy 77.37%, 72.32%, 70.01% on MUTAG, PROTEINS, NCI1 respectively, DGL-GCN obtained accuracy of 76.42%, 71.51%, and 67.63% respectively, while DGL-GCN only needs around half of the time and memory cost required by GCN.
>
> As for our previous experiment setup, there is a multi-GPU setup, in our experiments, we assign each layer to one GPU. When the number of GPUs is limited and the number of layers is large, it’s also ok to decouple the GNN into blocks (instead of layers), and assign each block to one GPU, the training process would be similar. In terms of convergence, the performance won’t be improved with more epochs.
>
> References:
>
> [1] Eugene Belilovsky, Michael Eickenberg, and Edouard Oyallon. Greedy layerwise learning can scale to imagenet. In International conference on machine learning, pp. 583–593. PMLR, 2019b.
>
> [2] Yuning You, Tianlong Chen, Zhangyang Wang, and Yang Shen. L2-gcn: Layer-wise and learned efficient training of graph convolutional networks. In Proceedings of the IEEE/CVF Conference on Computer Vision and Pattern Recognition, pp. 2127–2135, 2020.
>
> [3] Xu, Keyulu, et al. "How powerful are graph neural networks?." arXiv preprint arXiv:1810.00826 (2018).

---

### Decision · Program_Chairs · 2021-01-07
**Final Decision**

**Decision:**

Reject

**Comment:**

In this paper, the authors propose a new layer-by-layer training approach for GNN in particular for a large graph. The proposed approach can be easily parallelizable and scale well to a large graph. Reviewers are concerned about the novelty of the approach and the lack of theoretical analysis, and it is not well addressed by the rebuttal. Therefore, this paper is below the acceptance threshold of ICLR. I encourage the authors to revise the paper based on the reviewer's comments and resubmit it to a future venue.